# A Chimeric Vaccine against Porcine Circovirus Type 2: Meta-Analysis of Comparative Clinical Trials

**DOI:** 10.3390/vaccines11030584

**Published:** 2023-03-03

**Authors:** Barbara Poulsen Nautrup, Ilse Van Vlaenderen, Martha A. Mellencamp

**Affiliations:** 1EAH-Consulting, 52064 Aachen, Germany; 2CHESS, 2820 Bonheiden, Belgium; 3Zoetis, Parsippany, NJ 07054, USA

**Keywords:** porcine circovirus type 2, vaccine, meta-analysis, experimental challenge study, environmental challenge study

## Abstract

This meta-analysis compared the efficacy of a chimeric vaccine against porcine circovirus type 2 (PCV2) containing the genotypes PCV2a+b (Fostera^®^ Gold PCV MH [FOS-G]), with commonly used vaccines being derived from genotype PCV2a, considering the following parameters: average daily gain (ADG), mortality and market classification as full value and cull. Data from seven hitherto unpublished comparative US field trials with FOS-G (two experimental challenges and five natural environmental studies) were provided by the manufacturer. A complementary literature review revealed a Korean study, which was considered separately in meta-analysis. Competitors were Circumvent^®^ PCV-M (CV) and Ingelvac Circoflex^®^ + Ingelvac Mycoflex^®^ (IC + IM) in the US, and Porcilis^®^ (POR) in Republic of Korea. Heterogeneity between experimental and environmental challenge studies in the US was not significant, justifying a combined analysis. Over the entire feeding period, ADG (11 comparisons), mortality (12 comparisons) and market classification were not significantly different between FOS-G and its competitor in the US setting. In the Korean study, however, ADG was higher in pigs vaccinated with FOS-G compared to POR, whereas mortality was not significantly different.

## 1. Introduction

Porcine circovirus type 2 (PCV2) is recognized as one of the most important pathogens of the pig population worldwide [1]. In contrast to its non-pathogenic variant PCV1, PCV2 causes a disease complex which was previously described as wasting syndrome [2]. Apart from this, also called post-weaning multi-systemic wasting syndrome, many PCV2-associated clinical conditions, such as respiratory syndrome, congenital tremors, enteritis, dermatitis, nephropathy and reproductive issues, were later grouped as porcine circovirus-associated diseases in North America. The clinical manifestations vary widely from subclinical infections to severe, deadly porcine circovirus-associated disease. It may be a sporadic individual animal diagnosis, but also manifest as severe herd problem [3].

PCV2 is divided into eight genotypes, including the previously described genotypes PCV2a to PCV2f and the new genotypes PCV2g and PCV2h [4]. Three genotypes (PCV2a, PCV2b, and PCV2d) have been shown to exhibit worldwide distribution. Since the discovery of PCV2, two major changes in prevalence (“genotype shifts”) have been observed: in the mid-2000s, the initially prevalent PCV2a genotype was replaced by the PCV2b genotype, associated with a purported increase in virulence [5]. A new PCV2d genotype has been identified a decade ago and has become a challenge and serious economic problem in pig production systems all over the world [4]. Because of its close similarity to PCV2b, this genotype was initially referred to as PCV2b mutant [6].

In 2006, the first commercial vaccines became available in North America, leading to a decrease in morbidity and improved production efficacy, being the single highest-selling prophylactic agents in porcine husbandry. These and all other PCV2 vaccines launched until 2018 were derived from genotype PCV2a [1]. A meta-analysis published in 2011 found a significant effect of PCV2 vaccination on average daily gain (ADG) in all production phases, without statistically significant differences between the vaccines considered in their analysis (Circovac^®^ (Merial, Duluth, GA, USA), Circumvent^®^ PCV (Intervet/Schering-Plough, Kenilworth, NJ, USA), Ingelvac Circoflex^®^ (Boehringer Ingelheim Vetmedica, St. Joseph, MO, USA), Suvaxyn^®^ PCV2 (Fort Dodge Animal Health, Fort Dodge, IA, USA)) [7]. A mixed treatment comparison was published in 2014 and the authors concluded that Fostera^®^ PCV MH (Zoetis, Parsippany-Troy Hills, NJ, USA) had the lowest probability to be ranked best, compared to the three other vaccines included (Circumvent^®^ PCV (Merck Animal Health, Madison, NJ, USA), Ingelvac Circoflex^®^, and Ingelvac Circovac^®^) [8]. The analysis and its conclusion were, however, not undisputable [9].

In 2018 a new vaccine was launched in the United States (US), including two genotypes: PCV2a and PCV2b (Fostera^®^ Gold PCV2 MH, Zoetis; in the following, referred to as FOS-G), thereby accounting for the observed shift of genotypes and the similarity between PCV2b and PCV2d. Recently, FOS-G was also introduced into the Asian market.

Zoetis has conducted a series of field trials in the US, investigating the efficacy of the new vaccine under experimental and environmental infection conditions. Results have not been published in journals yet.

The aim of the present study was to evaluate the comparative effectiveness of FOS-G, using meta-analytic techniques. Our hypothesis was that FOS-G is as effective as other commonly used vaccines against PCV2.

## 2. Materials and Methods

### 2.1. Data Source and Data Collection

Results from the completed US field trials had not been published in scientific journals yet. The data were provided by Zoetis and were generated from previously conducted research projects. The various studies followed established standards for animal care and their use, including animal care and use committee approval at contract research organizations; following Pork Quality Assurance guidelines at customer sites as well as being conducted with swine at facilities operated under routine management practices in accordance with 7 U.S.C. 54; or were performed in accordance with the Guide for the Care and Use of Agricultural Animals in Research and Teaching.

Although the primary aim was to conduct a meta-analysis of results from hitherto unpublished studies, structured literature searches were conducted in PubMed and CAB Abstracts in order to identify other relevant studies. Search terms for PubMed were: (“circovirus”[MeSH Terms] OR “circovirus”[All Fields] OR (“porcine”[All Fields] AND “circovirus”[All Fields]) OR “porcine circovirus”[All Fields]) AND (“vaccines”[MeSH Terms] OR “vaccines”[All Fields] OR “vaccine”[All Fields]), and search terms used in CAB Abstract database were: circovirus AND porcine AND vaccine. Search years were 2015 until day of last search (1 April 2022). Additionally, the open internet (Google and Google Scholar) and websites of manufacturers of PCV2 vaccines were searched (Boehringer Ingelheim, Merck, MSD, Pharmgate, and Zoetis) for any eligible study.

Studies were defined as eligible which reported the efficacy of FOS-G compared to other PCV2 vaccines. The location was not restricted to the US. Both environmental infection studies, i.e., with natural exposure to PCV2, and experimental challenge studies qualified for inclusion.

A template for data collection was developed a priori in order to compile the following information: report number or reference; study location; type of challenge (experimental versus environmental); competitor vaccine; as well as the following outcome data: ADG; mortality; and market classification as full value or cull.

#### 2.1.1. Average Daily Gain

Effect size for ADG was the raw mean difference between FOS-G and competitor, being the most intuitive outcome. ADG was analyzed for the entire feeding period, i.e., from allocation to end of finisher (overall ADG), as well as separately for the nursery and finisher period. ADG was calculated for all remaining pigs at the end of each period. If not reported in the paper, the sample sizes for the nursery and finisher periods were estimated from the number of enrolled pigs and the number of dead and removals per period.

#### 2.1.2. Mortality

Effect size for mortality was the risk ratio between FOS-G and competitor vaccines. Similarly to the ADG, the risk of death was calculated over the entire feeding period (overall mortality), as well as for the nursery and finisher periods, considering the number of pigs present at the start of each period as a denominator in the risk calculation. For the analysis over the entire feeding period and nursery period, this was the number of enrolled pigs; for the finisher period, it was the number of pigs still alive after nursery. Removals were counted as deaths if not sold on the market.

#### 2.1.3. Market Classification as Full Value or Cull

Effect size for both market classifications was the risk ratio between FOS-G and competitor. The percentage of full value pigs was calculated from all enrolled pigs, whereas the percentage of culls was calculated from all pigs marketed.

### 2.2. Statistical Analyses

#### 2.2.1. Sequence of Analyses

Because of the presence of a Korean study being conducted in a pork production setting different from the US, the following sequence of analyses was defined: in primary analysis, heterogeneity between experimental and environmental US field trials was estimated. In case of no significant heterogeneity between the two subgroups, a combined analysis was run to estimate the overall outcome for the US setting. This combined outcome was compared with the overall outcome of the non-US comparisons by determining the heterogeneity between the two groups. In secondary analyses, subgroup analyses were used to evaluate if outcomes were different when considering the three competitors separately.

#### 2.2.2. Meta-Analysis

A meta-analysis was conducted using the statistical software CMA (Version 2.2, Biostat, Englewood, NJ, USA). For the combined analyses of subgroups, a mixed effect model was used as generally recommended. Heterogeneity between subgroups was quantified using the *Q*-test for heterogeneity, thereby testing the null hypothesis that all subgroups reveal the same effect. For each summary effect size, the *Z* statistic and corresponding *p*-value was used to determine if differences between FOS-G and competitor vaccines were statistically significant. To evaluate heterogeneity among studies (rather than subgroups), the *I*-squared statistic was used, which estimates the percentage of total variation across studies due to true heterogeneity rather than chance, thereby not being sensitive to the number of studies included [10]. *I*-squared values of 25%, 50% and 75% can be considered to be low, moderate and high heterogeneity, respectively [11]. Statistical significance was declared based on two-tailed tests at *p* < 0.05, except for the Q-test of heterogeneity, where a significance level of *p* < 0.10 is recommended to account for its low sensitivity [12].

A potential publication or selection bias was evaluated by creating a funnel plot of the log risk ratio versus standard error [13] for the parameter with the highest number of comparisons included. The trim and fill approach [14] was used, firstly, to assess the presence of any asymmetry (indicating bias) and, secondly, to provide the best estimate of unbiased outcome by recalculating the effect size until the tunnel plot was symmetric.

## 3. Results

### 3.1. Data Collection

Zoetis provided the final reports of seven field trials (16PRGBIO-01-01, 17PPTBIO-01-01, 17PPTBIO-01-02, 18PPTBIO-01-03, 18PPTBIO-01-04, 18PPTBIO-01-07, 18PPTBIO-01-08), which had not been published in journals at the time of our study. All trials were conducted in the US. Two studies used an experimental challenge protocol, whereas viral challenge was natural (environmental) in the remaining five studies. Competitors were Circumvent^®^ PCV-M (CV) and Ingelvac Circoflex^®^ + Ingelvac Mycoflex^®^ (IC + IM). Although the primary aim of all trials was to compare the efficacy against PCV2, all pigs in field trials were simultaneously vaccinated against *Mycoplasma hyopneumoniae*, as FOS-G and CV were combination products with antigens against both pathogens, and IC was mixed with IM (in accordance with its license in the US).

PubMed and CAB Abstract searches produced 375 and 466 references, respectively. In both databases, the same article has been identified reporting the comparative efficacy of FOS-G versus Porcilis^®^ PCV M Hyo (POR (MSD Animal Health, Boxmeer, Netherlands)) in naturally infected Korean pigs [15].

The comprehensive Google search, using numerous different search terms, as well as the search of manufacturers’ websites, revealed three brochures, sponsored by Zoetis and reporting parts of the results from three out of the seven provided field trials: 16PRGBIO-01-01 [16,17,18], 17PPTBIO-01-01 and 17PPTBIO-01-02 [17].

Accordingly, the seven US field trials provided by Zoetis, as well as the published Korean study, were eligible for meta-analysis. One of the seven US studies allowed the consideration of six comparisons (three study arms for FOS-G and two competitors) and the Korean study was the basis for two comparisons (two study arms for FOS-G and one competitor).

Because the field trials had not been published in peer-reviewed journals yet, a brief overview of the studies is given in Table 1 (experimental challenge studies) and Table 2 (environmental infection studies), presenting the outcomes as relevant for our meta-analyses. For reasons of completeness, relevant outcomes of the Korean study were included in Table 2 as well. Two experimental challenge studies (16PRGBIO01 and 18PPTBIO-01-03), as well as the Korean study, included a control arm (no vaccine). As the aim of this study was to investigate the chimeric vaccine’s comparative efficacy, these control arms were not considered in our analyses.

### 3.2. Meta-Analysis

#### 3.2.1. Average Daily Gain

Over the entire feeding period, the impact on ADG was not significantly different between FOS-G and competitors in US field trials, neither in the subgroup of experimental nor environmental challenge studies. No significant heterogeneity was observed between the two subgroups, thus allowing a combined analysis, resulting in a similar ADG between FOS-G and competitor (mean difference 0.96 g/day; *p* = 0.60). In the two comparisons of the Korean study, however, ADG was significantly higher in pigs vaccinated with FOS-G compared to those vaccinated with POR (mean difference 7.74 g/day; *p* < 0.001). In secondary analysis, no significant heterogeneity (*p* = 0.46) in outcome was found between the two competitors in US field studies (IC + IM and CV), whereas heterogeneity was significant (*p* = 0.01) if estimated for the three subgroups of competitor (IC + IM, CV, POR).

In the nursery period, ADG was significantly higher in US environmental infection studies. As heterogeneity was not significant between the subgroups of experimental and environmental challenge studies, a combined analysis was justified. Over all US studies, ADG was significantly higher (mean difference 2.46 g/day; *p* = 0.02) in pigs vaccinated with FOS-G versus competitor vaccines. The same applies to the two comparisons from the Korean study (mean difference 5.42 g/day; *p* = 0.049), resulting in no relevant heterogeneity between the studies of different origin. When analyzing the outcomes separately for the three vaccines, ADG was numerically higher in all comparisons and statistical significance was reached in the comparison versus CV and POR. In the finisher period, there were no significant differences in the US field trials between FOS-G and competitor vaccines, in any subgroup nor in the combined analysis. In the Korean study, ADG was significantly higher in pigs vaccinated with FOS-G compared to those vaccinated with POR (Table 3).

#### 3.2.2. Mortality

In the US field trials, mortality was not significantly different between FOS-G and competitors, neither in combined analysis (RR = 1.03; *p* = 0.70) nor in any of the subgroup analyses. This was true for all periods analyzed (overall, nursery, finisher). Although the overall risk of mortality was numerically lower over the entire feeding and finisher period in pigs vaccinated with FOS-G in the Korean study, results were not statistically significant (combined analysis: RR = 0.60; *p* = 0.39). No significant heterogeneity between subgroups could be found (Table 3).

#### 3.2.3. Market Classification

Only US field trials reported the percentage of pigs reaching full market value or being classified as cull. No significant differences were found between pigs vaccinated with FOS-G or competitor, neither in experimental nor environmental challenge studies. The same applied to the combined analyses, where the relative risk to be classified as full market or cull was 1.00 (*p* = 1.00) and 0.88 (*p* = 0.20), respectively, for pigs vaccinated with FOS-G. Again, no differences were observed when analyzing the outcome separately for the two competitors. No significant heterogeneity was observed in any subgroup analysis (Table 4).

#### 3.2.4. Publication and Selection Bias

The risk of publication or selection bias was estimated by creating a funnel plot for overall mortality, which was the parameter with the highest number of comparisons included. The trim and fill approach was run twice, once for the US field studies and once for all studies (US field trials and Korean study). A small asymmetry was estimated to the left (US field studies only) or to the right (all studies). When imputing one study in each analysis to make the funnel plots symmetric, the point estimates changed marginally, recognizable only at the second decimal place (Figure 1), which is indicative for a negligible publication or selection bias.

## 4. Discussion

We aimed to estimate the comparative efficacy of the chimeric vaccine FOS-G. Although the primary aim of our study was to estimate overall effect sizes from hitherto unpublished US field trials, other published studies could not be ignored for an unbiased assessment. Therefore, a literature review was performed, which revealed one study conducted in the Republic of Korea.

Our hypothesis was that FOS-G is similarly effective as other commonly used PCV2-vaccines. To our knowledge, this is the first meta-analysis considering FOS-G. Previous meta- or mixed-treatment analyses included the predecessor vaccine of FOS-G, namely Fostera^®^ PCV MH, and therefore results cannot be compared, especially as the inclusion of a second strain of PCV2, i.e., PCV2a + b, was intended to improve the efficacy of FOS-G compared to the previous vaccine.

The outcomes of our analysis confirmed our hypothesis. We can rationally state that FOS-G is at least as effective as competitor vaccines with respect to ADG and mortality, which can be regarded as key aspects of profitability in pork production [19]. No relevant differences were found for the market classification of pigs.

In the Korean study, pigs vaccinated with FOS-G had a significantly higher ADG compared to POR in all feeding periods. However, further evidence is needed to clarify if the better ADG observed in the Korean setting can be extrapolated to other countries and other pork production systems. The significantly higher ADG in the nursery period of the US field trials must be interpreted with caution, as the only study reporting a better ADG with the competitor vaccine had outcomes recorded for the overall feeding period only. Accordingly, results from this trial were missing for the nursery and finisher periods. Regardless, we can rationally state that FOS-G is at least as effective as the competitor vaccines used in the studies.

In secondary subgroup analysis, the impact of the competitor on differences versus FOS-G was evaluated (CV or IC + IM in US field trials and POR in Korean comparisons). In most analyses, the comparative efficacy of FOS-G was not significantly different from CV and IC + IM. It cannot be concluded from our analyses if the better outcome of ADG versus POR in all feeding periods or versus CV in the nursery period is attributable to the vaccine. Both vaccines, CV and POR, are different preparations of the same core vaccine, which are—with the exception of the Republic of Korea—not distributed in the same countries [20]. Further studies are warranted in this respect.

Results from our study cannot be extrapolated to competitor vaccines, which were not included in this meta-analysis. It should be noted, however, that CV and IC + IM were ranked highest regarding the improvement in ADG in the previous mixed treatment comparison of PCV2 vaccines [8].

As fundamental differences could not be ruled out between experimental challenges and environmental infections within the US field studies, the heterogeneity between the two groups had to be determined before the US field trials could be analyzed, combined or pooled in subgroup analyses, evaluating the impact of individual competitors. In all primary analyses, heterogeneity between experimental and environmental challenge studies was not statistically significant, even when applying a higher level of *p* > 0.10, as recommended to account for the low sensitivity of the Q-test for heterogeneity [12]. Therefore, we considered the combined analysis and pooling of environmental and experimental challenge US studies in secondary subgroup analysis as justified.

The US field studies and the two Korean comparisons were analyzed separately, thereby accounting for the different pork production systems and competitors. We abstained from calculating an overall effect size (including all studies), thereby following the recommendations in case of substantial dispersion [21].

Heterogeneity between studies (within groups) was estimated by *I*-squared statistics. The percentage of heterogeneity between studies that could not be explained by chance was zero in 24 of 40 different tests (the overall analysis of the two Korean comparisons also represented the subgroup of the competitor POR and was not counted twice) and no test revealed a high heterogeneity (≥75%). The missing or low to moderate heterogeneity might be regarded as an unexpected finding, as it also applied to combined analyses, but can be explained by the fact that our meta-analyses considered mean differences or risk ratios rather than absolute values, thereby adjusting for different baseline values, as also reported in previous meta-analysis [7].

Previous meta- or mixed-treatment analyses [7,8] included only naturally infected pigs. With the additional consideration of experimental challenge studies, we did not only increase the number of comparisons and thus statistical power, but also evaluated the comparative efficacy of the chimeric vaccine FOS-G in case of high infection pressure.

Two of the US field trials, as well as the Korean study, included a control group (unvaccinated pigs). We have not considered negative controlled trials in our meta-analyses, as a significant effect of FOS-G compared to no vaccination could be expected, being mandatory for licensing.

Our study has limitations. First, the results of the field trials conducted in the US had not been published in peer-reviewed journals at the time of study but were provided by the manufacturer of the chimeric vaccine (Zoetis). Accordingly, the source of data poses a risk of selection bias. In order to address this risk, the trim and fill approach [14] was used to estimate any asymmetry that indicates a systematic bias within the US trials. For reasons of completeness, the trim and fill approach was run a second time, also including the Korean study. The analyses were run for the overall mortality, as it was the parameter with the highest numbers of competitors. The approach revealed a minor asymmetry of one study in both analyses. When imputing the assumed missing study, the change in point estimate was marginally, i.e., not recognizable at the first decimal place. Therefore, we can rationally conclude that a publication or selection bias is of no concern.

The literature search was not a systematic review in its strictest sense. However, we performed a very comprehensive search, including PubMed and CAB abstracts, as well as the open internet and Google Scholar. Additionally, the trim and fill approach could not find a relevant publication bias. Although the method cannot determine if studies are actually missing, it can reveal a systematic bias in the used sample of studies. In other words, if the studies included in a meta-analysis are a random subset of all relevant studies, the failure to include missing studies will have no systematic impact on the effect size and the publication or selection bias is of no concern. Therefore, we believe that our approach can be regarded as a useful compromise to prevent a substantial time lag between a search and ultimate publication of the results, which has been described as a common drawback of systematic reviews [22].

The number of comparisons included in the analyses of market classification was small, thus limiting the transferability of results. Additionally, the definition of culls differed between studies. We used, however, the relative risk between the two vaccines (FOS-G and competitor) rather than absolute values, thereby adjusting for different base line values and reducing the impact of varying definitions. Additionally, the results of the analyses of full value pigs were very uniform. Therefore, we believe that our conclusions are still valid.

A further limitation relates to the dosage regimens used in the studies. While FOS-G, CV and POR are licensed for single use (2 mL at ≥3 weeks of age) and a two-dose regimen (1 dose of 1 mL at ≥3 days of age and another 1 mL 3 weeks later), IC + IM is licensed for single use at ≥3 weeks of age only. In the US field studies and the Korean studies, different dosage regimens were used for FOS-G. We did not analyze the impact of the different dosage regimens of FOS-G, as the number of comparisons in each subgroup did not qualify for meta-analyses. Further studies are needed to evaluate potential differences with different dosage regimens.

We conclude that FOS-G is at least as effective as the competitor vaccines included in this meta-analysis when considering the impact on ADG and is similarly effective with respect to the other production parameters considered. Our results are robust, as shown in different subgroup analyses, while lacking a relevant risk of publication or selection bias.

## 5. Conclusions

The chimeric vaccine against PCV2, Fostera^®^ Gold PCV MH, includes two genotypes (PCV2a+b), in order to adjust for shifts in genotype observed in recent years. In meta-analysis, Fostera^®^ Gold PCV MH was shown to be at least as effective as the competitor vaccines, and results were robust and valid for different levels of infection pressure.

## Figures and Tables

**Figure 1 vaccines-11-00584-f001:**
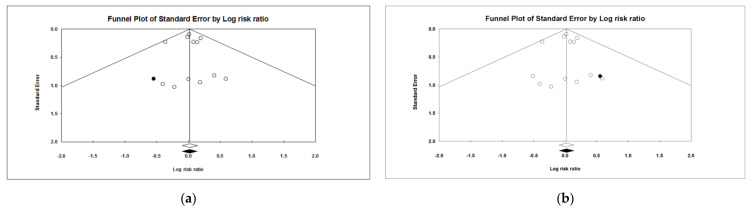
Funnel plot for assessing publication/selection bias, displaying the log risk ratio of overall mortality (from allocation to end of finisher) between Fostera^®^ Gold PCV MH and comparator (x-axis) by standard error (y-axis). The vertical line represents the overall effect size estimate. Without bias, the number of studies at both sides of the vertical line is expected to be equal. The “trim and fill” funnel plots display the observed (open circles) and the imputed study (filled circle), as well as the observed (open diamond) and imputed point estimate (filled diamond). (**a**): Funnel plot for US field studies only. (**b**) Funnel plot for all studies (US field studies and Korean studies).

**Table 1 vaccines-11-00584-t001:** Overview of outcome data that were included in meta-analysis and have been derived from hitherto unpublished US field trials, considering an experimental challenge model with porcine circovirus type 2.

	Average Daily Gain						Mortality						Full Value		Culls	
	Overall		Nursery		Finisher		Overall		Nursery		Finisher					
Vaccine (Doses)	N ^1^	Mean (SD)	N ^1^	Mean (SD)	N ^1^	Mean (SD)	N ^2^	Dead	N ^2^	Dead	N ^2^	Dead	N ^3^	Full Value	N ^3^	Culls
**Study: 16PRGBIO-01-01 (US)**																
FOS-G (D1)	77	731.64 (70.31)	97	454.05 (61.69)	72	993.37 (86.18)	110	9	NR		NR		NR		NR	
FOS-G (D2a)	83	695.36 (91.63)	104	434.54 (77.56)	81	936.21 (121.56)	110	4	NR		NR		NR		NR	
FOS-G (D2b)	83	722.12 (59.87)	98	452.68 (60.78)	79	980.21 (97.07)	110	6	NR		NR		NR		NR	
CV (D2a)	80	709.42 (65.77)	103	441.35 (64.86)	80	962.07 (92.53)	110	5	NR		NR		NR		NR	
IC + IM (D1)	80	713.95 (71.67)	102	444.07 (68.49)	79	961.16 (108.86)	110	6	NR		NR		NR		NR	
**Study: 18PPTBIO-01-03 (US)**																
FOS-G (D1)	697	793.79 (72.57)	NR		NR		750	39	NR		NR		750	692	711	4
IC + IM (D1)	703	802.86 (72.57)	NR		NR		750	36	NR		NR		750	698	714	0

FOS-G = Fostera^®^ Gold PCV MH (Zoetis); CV = Circumvent^®^ PCV-M (Merck Animal Health); IC + IM = Ingelvac Circoflex^®^ + Ingelvac Mycoflex^®^ (Boehringer Ingelheim Animal Health). D1 = 1 dose of 2 mL vaccine at 3 weeks of age; D2a = 2 doses of 1 mL vaccine each at 3 days of age and 3 weeks later; D2b = 2 doses of 1 mL vaccine each at 3 and 6 weeks of age; NR = not reported. ^1^ number of pigs considered for the calculation of average daily gain; ^2^ number of pigs enrolled; ^3^ number of pigs marketed.

**Table 2 vaccines-11-00584-t002:** Overview of outcome data that were included in meta-analysis and have been derived from hitherto unpublished US field trials and a Korean study, considering natural infection with porcine circovirus type 2.

	Average Daily Gain (g/day)						Mortality						Full Value		Culls	
	Overall		Nursery		Finisher		Overall		Nursery		Finisher					
Vaccine (Doses)	N ^1^	Mean (SD)	N ^1^	Mean (SD)	N ^1^	Mean (SD)	N ^2^	Dead	N ^2^	Dead	N ^2^	Dead	N ^3^	Full Value	N ^3^	Culls
**Study: 17PPTBIO-01-01 (US)**																
FOS-G (D2b)	582	736.63 (77.11)	596	369.22 (77.11)	579	904.01 (99.79)	618	31	618	19	599	12	618	578	581	3
CV (D2b)	556	734.82 (81.65)	574	369.68 (81.65)	547	900.83 (104.33)	623	45	623	26	597	19	623	565	571	6
**Study: 17PPTBIO-01-02 (US)**																
FOS-G (D1)	1262	752.96 (81.65)	1296	353.80 (68.04)	1257	920.79 (99.79)	1322	39	1322	12	1310	27	1322	588	1261	82
CV (D1)	1267	757.50 (81.65)	1307	353.80 (72.57)	1265	925.33 (99.79)	1324	34	1324	12	1312	22	1324	611	1268	93
**Study: 18PPTBIO-01-04 (US)**																
FOS-G (D1)	1064	698.53 (91.63)	1145	392.81 (85.28)	1064	845.04 (119.29)	1172	96	1172	27	1145	69	1172	1022	1042	20
IC + IM (D1)	1072	696.72 (98.88)	1137	387.82 (88.00)	1072	845.04 (126.10)	1177	98	1177	41	1136	57	1177	1014	1042	28
Study: 18PPTBIO-01-07 (US)																
FOS-G (D2b)	NR		NR		NR		2775	211	NR		NR		NR		2443	61
CV (D2b)	NR		NR		NR		2774	207	NR		NR		NR		2428	66
**Study: 18PPTBIO-01-08 (US)**																
FOS-G (D2b)	923	799.68 (18.14)	994	300.28 (22.68)	923	979.31 (23.59)	1042	85	NR		NR		1042	938	940	2
CV (D2b)	850	796.51 (19.50)	902	297.56 (29.03)	850	974.77 (22.23)	937	63	NR		NR		937	857	858	1
**Um et al., 2021 (Republic of Korea)** [15]																
FOS-G (D1)	117	656.06 (11.85)	119	399.90 (25.44)	117	775.62 (20.65)	120	3	120	1	119	2	NR		NR	
FOS-G (D2b)	117	654.74 (11.38)	119	401.89 (24.05)	117	772.73 (18.45)	120	3	120	1	119	2	NR		NR	
POR (D1)	115	647.65 (14.17)	119	395.51 (24.20)	115	765.23 (22.73)	120	5	120	1	119	4	NR		NR	

FOS-G = Fostera^®^ Gold PCV MH (Zoetis); CV = Circumvent^®^ PCV-M (Merck Animal Health); IC + IM = Ingelvac Circoflex^®^ + Ingelvac Mycoflex^®^ (Boehringer Ingelheim Animal Health). D1 = 1 dose of 2 mL vaccine at 3 weeks of age; D2b = 2 doses of 1 mL vaccine each at 3 and 6 weeks of age; NR = not reported. ^1^ number of pigs considered for the calculation of average daily gain; ^2^ number of pigs enrolled; ^3^ number of pigs marketed.

**Table 3 vaccines-11-00584-t003:** Results of meta-analyses, evaluating the comparative efficacy of Fostera^®^ Gold PCV MH (FOS-G, Zoetis) on average daily gain and mortality in US field trials and a Korean study.

		Average Daily Gain, g/day				Mortality				
Analysis	n	Mean Difference	*p*-Value	*I*^2^, %	*p*-Value between *	n	Risk Ratio	*p*-Value	*I^2^*, %	*p*-Value between *
**Overall (from allocation until end of finisher)**										
**Primary analysis (1st subgroup and combined analysis)**										
Experimental challenge (US)	7	−0.616	0.914	22.99	0.770	7	1.098	0.629	0.00	0.710
Environmental challenge (US)	4	1.136	0.551	43.45		5	1.017	0.808	8.90	
**Combined**	**11**	**0.960**	**0.595**	**45.19**	0.003	**12**	**1.026**	**0.695**	**0.00**	0.370
Um et al., 2021 (Republic of Korea) [15]	2	7.735	<0.001	0.00		2	0.600	0.391	0.00	
**Secondary analysis (2nd subgroup analysis)**										
FOS-G versus IC + IM (US)	5	−2.406	0.579	35.18	0.461 ^†^	5	1.012	0.919	0.00	0.889 ^†^
FOS-G versus CV (US)	6	1.216	0.599	34.62		7	1.031	0.672	0.00	
FOS-G versus POR (Republic of Korea)	2	7.735	<0.001	0.00	0.010 ^‡^	2	0.600	0.391	0.00	0.663 ^‡^
**Nursery**										
**Primary analysis (1st subgroup and combined analysis)**										
Experimental challenge (US)	6	5.466	0.364	0.00	0.613	NR				NA
Environmental challenge (US)	4	2.373	0.020	0.00		3	0.738	0.075	0.00	
**Combined**	**10**	**2.459**	**0.015**	**0.00**	0.313	**3**	**0.738**	**0.075**	**0.00**	0.805
Um et al., 2021 (Republic of Korea) [15]	2	5.423	0.049	0.00		2	1.000	1.000	0.00	
Secondary analysis (2nd subgroup analysis)										
FOS-G versus IC + IM (US)	4	4.848	0.147	0.00	0.454 ^†^	1	0.661	0.091	NA	0.531 ^†^
FOS-G versus CV (US)	6	2.221	0.035	0.00		2	0.820	0.406	0.00	
FOS-G versus POR (Republic of Korea)	2	5.423	0.049	0.00	0.454 ^‡^	2	1.000	1.000	0.00	0.797 ^‡^
**Finisher**										
**Primary analysis (1st subgroup and combined analysis)**										
Experimental challenge (US)	6	11.932	0.249	0.00	0.334	NR				NA
Environmental challenge (US)	4	1.665	0.491	44.27		3	1.070	0.699	27.44	
**Combined**	**10**	**2.194**	**0.351**	**18.20**	0.041	**3**	**1.070**	**0.699**	**27.44**	0.291
Um et al., 2021 (Republic of Korea) [15]	2	8.847	<0.001	0.00		2	0.500	0.321	0.00	
**Secondary analysis (2nd subgroup analysis)**										
FOS-G versus IC + IM (US)	4	1.321	0.792	0.00	0.894 ^†^	1	1.201	0.293	NA	0.467 ^†^
FOS-G versus CV (US)	6	2.098	0.481	36.17		2	0.914	0.787	52.31	
FOS-G versus POR (Republic of Korea)	2	8.847	<0.001	0.00	0.134 ^‡^	2	0.500	0.321	0.00	0.399 ^‡^

FOS-G = Fostera^®^ Gold PCV MH (Zoetis); IC + IM = Ingelvac Circoflex^®^ + Ingelvac Mycoflex^®^ (Boehringer Ingelheim Animal Health); CV = Circumvent^®^ PCV-M (Merck Animal Health); POR = Porcilis^®^ PCV M Hyo (MSD Animal Health); n = number of comparisons; NA = not applicable; NR = not reported. * estimates heterogeneity between subgroups; ^†^ heterogeneity between 2 subgroups of comparator; ^‡^ heterogeneity between three subgroups of comparator.

**Table 4 vaccines-11-00584-t004:** Results of meta-analyses, evaluating the comparative efficacy of Fostera^®^ Gold PCV MH (FOS-G, Zoetis) on the relative risk of being classified as full market pigs or culls.

Market Classification		Full Value				Culls				
	n	RR	*p*-Value	*I*^2^, %	*p*-Value between *	n	RR	*p*-Value	*I^2^*, %	*p*-Value between *
**Primary analysis (1st subgroup and combined analysis)**										
Experimental challenge	1	0.991	0.552	NA	0.504	1	9.038	0.139	NA	0.116
Environmental challenge	4	1.004	0.736	47.17		5	0.867	0.166	0.00	
**Combined**	**5**	**0.999**	**0.885**	**36.44**	**NA**	**6**	**0.876**	**0.200**	**0.00**	**NA**
**Secondary analysis (2nd subgroup analysis)**										
FOS-G versus IC + IM	2	1.001	0.954	0.00	0.971	2	1.668	0.669	64.24	0.602
FOS-G versus CV	3	1.000	0.992	62.82		4	0.891	0.298	0.00	

FOS-G = Fostera^®^ Gold PCV MH (Zoetis); IC + IM = Ingelvac Circoflex^®^ + Ingelvac Mycoflex^®^ (Boehringer Ingelheim Animal Health); CV = Circumvent^®^ PCV-M (Merck Animal Health); n = number of comparisons; RR = risk ratio; NA = not applicable. * estimates heterogeneity between subgroups.

## Data Availability

The data supporting the conclusions of this work are included within the article. The original data will not be shared as these are the property of the study sponsor.

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
