# Peer review of "A Chimeric Vaccine against Porcine Circovirus Type 2: Meta-Analysis of Comparative Clinical Trials"

_vaccines, 2023, doi:10.3390/vaccines11030584_

Round 1

Reviewer 1 Report

This manuscript reports a meta-analysis of porcine circovirus vaccines, focusing on the comparison of a newly developed chimeric vaccine against circovirus type 2 with  a number of other vaccines regarding the average daily gain and mortality as main outcome parameters. Also the relative risk of being classified as full market pigs or culls was analysed. All studies were field studies in the USA, except for one study from South Korea, which also was different regarding the study population: therefore, the analysis was with respect of this study was done separately. The separate meta-analyses revealed no significant differences regarding mortality but regarding average daily gain there was a significantly higher gain in the Korean study. Since the other studies in the comparison were field trials in the USA, an unequivocal conclusion about this difference in gain cannot be made.

In view of the relevance of circovirus vaccines for animal health in the pork industry the conduct and outcome of this meta-analysis is quite relevant. The study design, data analysis, and description and interpretation of the results is well presented. This also includes the presentation of study limitations and the rationale of considering the Korean study separately.

Noteworthy, the newly developed chimeric vaccine is from the same company as one of the authors, and this company funded the study. This is correctly described in the conflict of interest statement.

There is one suggestion in a minor revision of the manuscript, namely the use of the word “efficacy” and related terms such as “effective”. This could create confusion because the outcome parameters do not include efficacy of the vaccines. This is illustrated by the forelast paragraph of the discussion describing the different dose levels and dosing protocols: this was not included in the study, and also not the efficacy of the vaccins in terms of prevention of infection and/or disease.

Also, it is noted that the first sentence of the conclusion is not a conclusion. Rather, this is a rationale for the conduct of this study

Reviewer 2 Report

This manuscript reported that the meta-analysis and the efficacy of a chimeric vaccine against porcine circovirus type 2 (PCV2a+b) with 10 used PCV2a vaccines. Over the entire feeding period, ADG (18 comparisons), mortality (12 comparisons) and market classification were not significantly different between FOS-G and competitor in the US setting. It contains some useful insights and is helpful for the prevention of porcine circovirus infections in the world. Therefore, its worth of publication on Vaccines in present form. 

Reviewer 3 Report

In the manuscript entitled “A chimeric vaccine against porcine circovirus type 2: Meta-analysis of comparative clinical trials”, Poulsen Nautrup et al. reported a comprehensive meta-analysis to evaluate the efficacy of a new chimeric vaccine against PCV2 that derived from genotype PCV2a and PCV2b. The results suggested that the vaccine is as effective as other commonly used vaccines in protecting pigs against PCV2 infection based on several practical parameters including average daily gain, mortality and full vs cull market value. The authors included unpublished manufacturer data from several field trials of both experimental and environmental challenges which are particularly desirable for the study. Overall, the manuscript is well written, concise, methodical and scientifically sound. The authors took on a systematic approach of data analysis and comparison, and also included a comprehensive search and review of existing literature. This article will be of interest to the readership of Vaccines including pig producers, field scientists, and vaccine manufacturers.
